# Transcriptome analysis of *Xanthomonas oryzae* pv. *oryzicola* exposed to $H_2O_2$ reveals horizontal gene transfer contributes to its oxidative stress response

Yuan Fang[1], Haoye Wang[1], Xia Liu[1], Dedong Xin[1], Yuchun Rao[1]*, Bo Zhu[2]*

**1** College of Chemistry and Life Sciences, Zhejiang Normal University, Jinhua, P.R. China, **2** School of Agriculture and Biology, Shanghai Jiao Tong University/Key Laboratory of Urban Agriculture by Ministry of Agriculture of China, Shanghai, China

* ryc@zjnu.cn (YR); bzhu1981@sjtu.edu.cn (BZ)

**Data Availability Statement:** All relevant data are within the manuscript and its Supporting Information files.

## Abstract

*Xanthomonas oryzae* pv. *oryzicola* (*Xoc*), the causal agent of bacterial leaf streak, is one of the most severe seed-borne bacterial diseases of rice. However, the molecular mechanisms underlying *Xoc* in response to oxidative stress are still unknown. In this study, we performed a time-course RNA-seq analysis on the *Xoc* in response to $H_2O_2$, aiming to reveal its oxidative response network. Overall, our RNA sequence analysis of *Xoc* revealed a significant global gene expression profile when it was exposed to $H_2O_2$. There were 7, 177, and 246 genes that were differentially regulated at the early, middle, and late stages after exposure, respectively. Three genes (*xoc_1643*, *xoc_1946*, *xoc_3249*) showing significantly different expression levels had proven relationships with oxidative stress response and pathogenesis. Moreover, a hypothetical protein (XOC_2868) showed significantly differential expression, and the *xoc_2868* mutants clearly displayed a greater $H_2O_2$ sensitivity and decreased pathogenicity than those of the wild-type. Gene localization and phylogeny analysis strongly suggests that this gene may have been horizontally transferred from a *Burkholderiaceae* ancestor. Our study not only provides a first glance of *Xoc*'s global response against oxidative stress, but also reveals the impact of horizontal gene transfer in the evolutionary history of *Xoc*.

## Introduction

*Xanthomonas* consists of a large number of phytopathogenic bacteria that infect more than 400 host species, including a wide variety of economically important plants, such as rice, citrus, banana, cabbage, and bean [1, 2]. One of them, *Xoc*, is the causal agent of bacterial leaf streak (BLS) of rice (*Oryza sativa*), and is considered one of the most destructive seed-borne bacterial diseases of rice in many Asian countries and parts of Africa [3]. The interaction complexity between *Xoc* and rice results from the outcome of a longstanding and ongoing evolutionary battle, in which the bacteria attempts to invade and multiply, while the rice plant's cells attempt to recognize and defend against this invasion. Of these plant responses, oxidative burst is one of the initial response against pathogens [4].

**Funding:** This work was supported by Zhejiang National Natural Science Foundation of China (LY18C140003), http://www.zjnsf.gov.cn/, to YF; National Natural Science Foundation of China (31200003), http://www.nsfc.gov.cn/, to BZ. The funders had no role in study design, data collection and analysis, decision to publish, or preparation of the manuscript.

**Competing interests:** The authors have declared that no competing interests exist.

Oxidative burst is a process in which high concentrations of reactive oxygen species (ROS) are produced at the plasma membrane in the vicinity of a pathogen [4, 5]. This activity in plants may directly kill the pathogen, slows its growth by producing toxins, or even act as a signaling cascade leading to various defenses, including the hypersensitive response (HR) and cell wall modifications [6]. Since ROS activity is a common feature of plant defense systems and the mechanism involved in pathogen cell death, any pathogen that is able to resist this effect is likely privileged. These mechanisms require complex and well-orchestrated reactions involving both radical scavenging and enzymatic repair activities [7]. The cellular response to oxidative stress in the model organism, *Escherichia coli*, has been largely elucidated. When *E. coli* cells are treated with a low dose of $H_2O_2$, growth arrest occurs immediately and the expression of approximately 30 genes is maximally induced within 10–30 minutes of treatment [8]. Among them, the oxidized form of the transcriptional regulator, OxyR, induces many genes such as *katG* (encoding catalase G), *ahpCF* (encoding alkyl hydroperoxide reductase), and *trxC* (encoding reduced thioredoxin) to remove intracellular $H_2O_2$, maintain redox homeostasis and ultimately enable cells to resume growth [9, 10]. These studies demonstate that the genes involved in oxidative-stress response are vital for bacteria survival. However, there is limite knowledge on oxidative response network of phytobacteria using time-course system analysis although it is important to elucidatethe molecualr mechanisms that plant pathogens use to resist oxidative stress.

RNA-seq is widely used to investigate differential gene expression in many bacteria [8]. Most of the oxidative stress research using this technology were carried out on human pathogens and environmental bacteria [11, 12]. The application of RNA-seq to *Xoc on* oxidative stress may provide new insights into molecular mechanisms on oxidative response network.

BLS256, the representative genome of *Xoc*, has been whole-genome sequenced [13]. Interestingly, more than 30% of the coding genes in this genome are hypothetical genes.

In this study, we identified differentially expressed genes in the *Xoc* strain BLS256 in response to a time-course $H_2O_2$ treatment (early, middle, and late) using the Illumina RNA-Seq platform. Furthermore, we discovered one hypothetical protein, in addition to the genes previously confirmed to be triggered by oxidative stress, which showed high-fold differential expression compared with the control. This finding strongly suggests that this gene could have been horizontally transferred from other microbe organism.

## Materials and methods

### Strains, plasmids and culture conditions

The bacterial strains and plasmids used in this study are listed in S1 Table. *Escherichia coli* strains were cultivated at 37 ˚C in Luria-Bertani (LB) medium or on LB agar plates. Unless otherwise specified, *Xoc* strains were grown at 28 ˚C in nutrient broth (NB) (0.1% yeast extract, 0.3% beef extract, 0.5% polypeptone, and 1% sucrose), nutrient agar (NA) (NB with 15 g $L^{-1}$ agar), NA without sucrose (NAN), NA with 10% sucrose (NAS), and NB without beef extract and sucrose (NY). In some experiments, strains were grown in MMX minimal medium [0.5% glucose, 0.2% $(NH_4)_2SO_4$, 0.1% trisodium, citrate dihydrate, 0.4% $K_2HPO_4$, 0.6% $KH_2PO_4$, 0.02% $MgSO_4 \cdot 7H_2O$]. Kanamycin (Km) and Ampicillin (Amp) were added in concentration of 50μg $mL^{-1}$ and 100μg $mL^{-1}$, respectively, as required.

### Oxidative stress treatments and total RNA preparation

$H_2O_2$ resistance assays were performed as described previously described [14] with some modifications. *Xoc* strains were cultured to the mid-log phase (OD 600 = 1.0 ~ 1.2) in NB medium. Cultures were treated with 0.1mM $H_2O_2$ (Fluka) and constantly stirred at 330 rpm in a 28 ˚C

shaking incubator. At time of 0, 7, 15, and 45 min, aliquots were withdrawn and pelleted by centrifugation at 4 ˚C. Cell The cell pellets was were washed two times with cold PBS and the total RNA immediately extracted using fthe RNeasy Protect Bacteria Mini Kit (QIAGEN) protocal. Two biological replicates were performed in this experiment.

## mRNA purification and cDNA synthesis

Ten micrograms from each total RNA sample were treated with the MICROBExpress Bacterial mRNA Enrichment kit and RiboMinus™ Transcriptome Isolation Kit (Bacteria) (Invitrogen). Bacterial mRNAs were chemically fragmented to the size range of 200–250 bp using 1 X fragmentation solution for 2.5 min at 94˚C. cDNA was generated according to instructions given in the SuperScript Double-Stranded cDNA Synthesis Kit (Invitrogen). Each mRNA sample was mixed with 100 pmol of random hexamers, incubated at 65 ˚C for 5 min, and chilled on ice for 2 min. 4 µL of First-Strand Reaction Buffer (Invitrogen), 2 µL of 0.1 M DTT, 1 µL of 10 mM RNase-free dNTPmix, 1 µL of SuperScript III reverse transcriptase (Invitrogen) were then added to each sample and incubated at 50 ˚C for 1 h. To generate the second strand, the following Invitrogen reagents were then added: 51.5 µL of RNase-free water, 20 µL of second-strand reaction buffer, 2.5 µL of 10 mM RNase-free dNTP mix, 50 U *E. coli* DNA Polymerase, and 5 U *E. coli* RNase H, and incubated at 16 ˚C for 2.5 h.

## Library construction and Illumina sequencing

The Illumina Paired End Sample Prep kit was used for RNA-Seq library creation according to the manufacturer's instructions. Fragmented cDNA was end-repaired, ligated to Illumina adaptors, and amplified by 18 cycles of PCR. Paired-end 100-bp reads were generated by high-throughput sequencing with the Illumina Hiseq2000 Genome Analyzer instrument.

## RNA-Seq data analysis

After removing the low quality reads and adaptors, pair-end reads were mapped to the reference genome BLS256 using Bowtie1.1.1 with default parameters [15]. If reads mapped to more than one location, then only the reading havingthe highest alignment score was kept. Reads mapping to rRNA and tRNA regions were removed from further analysis. Four time-points samples were prepared in this study: 0min, 7min, 15min and 45min, defined as WT, early, middle and late stage, respectively. All samples were compared with the 0 min samples to detect the differential expressed genes (DEGs). After obtaining the number of reads for every sample, the edgeR with TMM normalization method [16] was used to determine the DEGs [17]. FDR value < 0.05 was selected as the cutoff for further analysis. Cluster 3.0 and TreeView were used to represent the cluster of DEGs over time-series samples [18].

## Quantitative real-time PCR

Bacterial total RNA was extracted following RNeasy Protect Bacteria Mini Kit (QIAGEN) instructions and used in generating the first strand complementary DNA (cDNA) per Takara PrimeScript RT reagent Kit with gDNA Eraser's (Takara) protocol. After detecting RNA quantity and quality by Nanodrop ND1000 spectrophotometer V 3.5.2 (NanoDrop Technologies, Wilmington, DE, USA), 1 µg of the total RNA was incubated at 42 ˚C for 20 mins to eliminate the gDNA by gDNA Eraser before obtaining cDNA. The reverse transcription reaction was accomplished by incubated at 37 ˚C for 15 min and then 85 ˚C for 5 s in the presence of random RT primers. The cDNA was then used directly as the template for qRT-PCR using a SYBR Green master mix (Applied Biosystems) on an ABI Prism 7000 sequence detection

system (Applied Biosystems). Normalized expression levels of the target gene transcripts were calculated relative to the rRNA using the ΔΔCT method, where CT is the threshold cycle [19]. Three biological replicates were carried out in this experiment.

## Construction of defective deletion mutant and complementation

To investigate the role of interested genes in *Xoc*, In-frame deletion mutations were constructed in BLS256 using homologous recombination. Briefly, two fragments flanking the left and right of corresponding genes were amplified from the wide-type genomic DNA with primer pairs listed in S2 Table. The amplified fragments were cloned into **pMD18-T** vectors (TaKaRa), confirmed by sequence analysis, and then digested and subcloned into vector **pKMS1** [20] at *BamH*I and *Pst*I (or *Sal*I) sites. The resultant recombinant plasmids were introduced into BLS256 by electroporation, and transformants were plated on NAN plates supplemented with kanamycin. Colonies resulting from a single homologous crossover (integration of deletion construct at either the left or right border of target gene) were then transferred to NBN broth, grown for 12 h at 28°C, and then plated on NAS plates for sucrose-positive deletion mutant selection. Sucrose resistant colonies were visible within 3 to 4 days and then transferred to NA plates and NA plus kanamycin plates. Since kanamycin-sensitive colonies could be mutants containing a second homologous crossover, these were further examined by PCR amplification with the primer pairs.

In order to complement the deletion mutants, the full-length of corresponding genes including promoter regions were amplified using primer pairs listed in S2 Table. The amplified DNA fragments were cloned into **pUFR034** vectors [20] at the *BamH*I and *Pst*I (or *Sal*I) sites to create the recombinant plasmids. The recombinant plasmids were transferred into corresponding mutants by electroporation, and transformants were screened on NA plates with kanamycin.

## H$_2$O$_2$ resistance assay

The assay plates were prepared by adding H$_2$O$_2$ to the sterilized NB medium in concentrations of 0, 0.1 and 0.25 mM, respectively. Strains were cultured to the mid-log phase (OD 600 = 1.0) in NB medium, and three-fold and nine-fold dilutions were made. 5-μL aliquots of the initial culture and diluted cultures for each strain were spotted onto NB agar plates (in triplicate) and cultured for 36 h at 28 °C [21].

## Pathogenicity assays

Bacteria were prepared based on previous reported method [21]. Briefly, *Xanthomonas* cells were grown in NB broth at 28°C and 200 rpm for 16 h, when cells approached the exponential phase of growth. Bacterial cells were then harvested by centrifugation, washed twice, and resuspended in sterile water to an optical density at 600 nm (OD600) of 0.3 (approximately 1×10$^8$ CFU/mL). Bacteria were inoculated into leaves of adult rice plants (*Oryza sativa* cv. IR24, susceptible to *Xoc*, 2 months old) using leaf piercing for lesion length measurement for evaluating water-soaked symptoms. All plants were maintained in a greenhouse as described previously [13]. Five leaves were inoculated for each independent experiment, and each treatment was repeated at least three times.

## Phylogenetic analysis

Interested gene was first compared against non-redundant database for homologs searching with E<0.0001 as cutoff. Next, 40% alignment similarity and 80% coverage parameter was

used to define the real homologue genes. MAFFT was used to generate the multiple sequences alignment [22]. Maximum Likelihood phylogeny was finally evaluated and built by PhyML [23] using a JTT model and a gamma distribution with eight rate categories. We performed 1,000 bootstraps to gain branch support values.

## Results

### Global overview of the *Xoc* transcriptome in response to $H_2O_2$

In this study, 100-bp paired-end deep sequencing was performed on all the tested samples. In general, more than 40M reads were generated from each single sample. After adaptor removal, quality control, and removal of reads, mapped to ribosomal RNA, 7M to 10M confident reads remained. The sequencing depth in this experiment was more than 150 X, which is sufficient for further statistical analysis. In general, 7, 177, and 246 genes were differentially regulated in the early, middle, and late stages, respectively (Fig 1, S3 Table). The overall number of differentially expressed genes (DEGs) was similar to the number found in recent research related to human pathogens and environmental bacteria [24, 25], with the exception of DEGs in the early state (7 min), which were almost 10 times smaller than in our study. This number, indicates the oxidative response difference that may occur between plant pathogen versus other bacteria.

All of the RNA-seq data has been deposited in BioProject database (https://www.ncbi.nlm.nih.gov/bioproject) and the accession number is PRJNA350867.

### General gene expression kinetics among the time-course samples

Previous research indicates that bacteria require dynamic regulatory networks at different time-points when they are exposed to environmental stress [24, 26]. To capture this variable activity, gene expression among the time-course samples was generated and results are shown in Fig 1. In general, clusters 1 and 2 had a decreased induction whereas clusters 4 and 5 had an increased induction through time (Fig 1). By contrast, the clusters 3 and 6 consisted of those genes whose expression peaked at the middle time stage.

Gene Ontolog (GO) and Kyoto Encyclopedia of Genes and Genomes (KEGG) enrichment analysis were carried out based on the STRING V10 database [27], with a $p < 0.01$ after a Bonferroni correction [28] set as the cutoff. The GO enrichment analysis revealed the categories of transport (GO:0006810), cell outer membrane (GO:0009279), and acyl-CoA dehydrogenase activity (GO:0003995) as significant regulation based on this cutoff for clusters 1 and 2 (Table 1). The regulation of cellular process (GO:0050794), chemotaxis (GO:0006935), single organism signaling (GO:0044700), response to external stimulus (GO:0009605), molecular transducer activity (GO:0060089), and signal transducer activity (GO:0004871) were significant for clusters 4 and 5 (Table 1). Membrane protein (GO:0016020), outer membrane protein (GO:0019867), and cellular component (GO:0005575) were significant for clusters 3 and 6 (Table 1). The KEGG enrichment results suggested that only valine, leucine, and isoleucine degradation was significant for clusters 1 and 2 (Table 1). Interestingly, we found many TonB-dependent receptors (TBDRs) that were differentially expressed in all the three time stages, indicating their importance in oxidative stress (S3 Table).

### Gene clusters involved in the DEGs

Since those genes forming a bacterial gene cluster are always involved in a common pathway or function [29], the DEGs located in the gene clusters we identified may play a fundamental role in oxidative stress. Based on our criterion—that at least three tandem genes must be

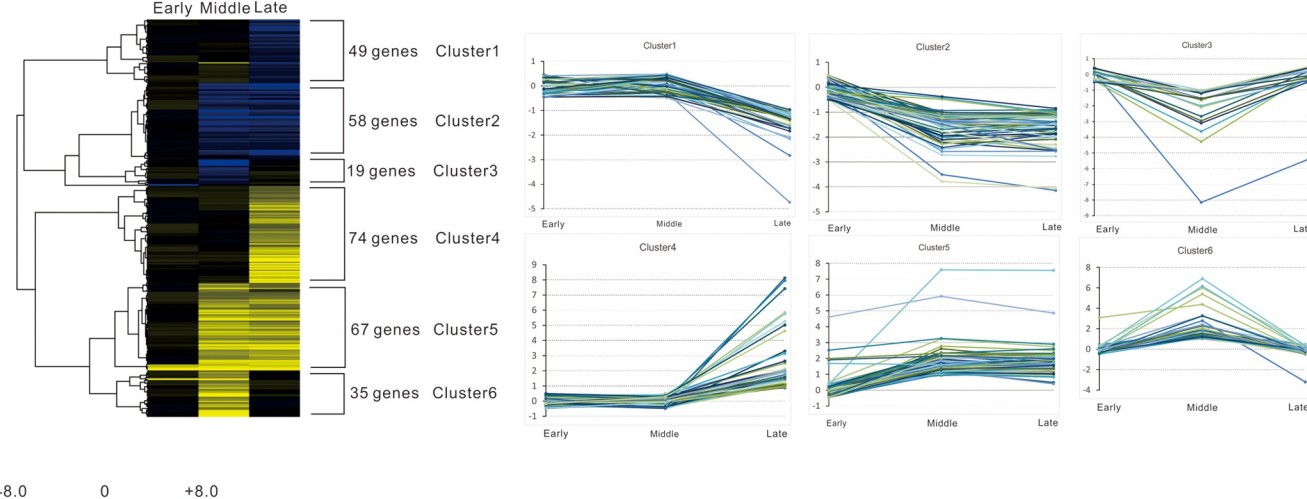

**Fig 1. Time-course transcriptome study of *Xoc* (*Xanthomonas oryzae* pv. *oryzicola*) resistance to an oxidative stress response.** Hundreds of genes are differentially expressed at different time points (early, middle, and late) during the $H_2O_2$ treatment when compared with the 0-min sample. These differentially expressed genes can be classified into different clusters based on their expression patterns. The line from -8.0 to 8.0 represents the $Log_2$ fold change value.

differentially expressed to constitute a gene cluster—we identified all the possible clusters and listed them in S4 Table. Not surprisingly, many reported oxidative stress-associated gene clusters were found involved, such as those for the alkyl hydroperoxide reductase, *suf* operon and *pst* operon [30, 31]. Furthermore, we also found a large gene cluster encoding ribosomal proteins that were up-regulated in our study (S3 Table). In addition, we found an F1F0 ATPase complex cluster that was up-regulated under oxidative stress.

**Table 1. The functional groups and their significance in different gene clusters.**

| Functional group | Clusters | | | *p* value[&] |
| --- | --- | --- | --- | --- |
| | 1 and 2 | 4 and 5 | 3 and 6 | |
| **GO category** | | | | |
| Transport (GO:0006810) | + | | | 6.50E-02 |
| Cell outer membrane (GO:0009279) | + | | | 3.13E-02 |
| Acyl-CoA dehydrogenase activity (GO:0003995) | + | | | 1.61E-03 |
| Regulation of cellular process (GO:0050794) | | + | | 1.97E-03 |
| Chemotaxis (GO:0006935) | | + | | 1.60E-02 |
| Single organism signaling (GO:0044700) | | + | | 5.32E-02 |
| Response to external stimulus (GO:0009605) | | + | | 9.49E-02 |
| Molecular transducer activity (GO:0060089) | | + | | 5.94E-03 |
| Signal transducer activity (GO:0004871) | | + | | 7.08E-02 |
| Membrane (GO:0016020) | | | + | 1.09E-03 |
| Outer membrane (GO:0019867) | | | + | 5.20E-02 |
| Cellular component (GO:0005575) | | | + | 7.46E-02 |
| KEGG category | | | | |
| Valine, leucine and isoleucine degradation | + | | | 4.06E-02 |

[&]: Bonferroni test

## Quantitative real time PCR experiments confirms the gene expression profiles

The RNA-seq results were validated with a qRT-PCR analysis of four selected genes (*xoc_1643*, *xoc_1946*, *xoc_2868*, and *xoc_3249*) that showed different levels of expression at 7, 15, and 45 minutes (S3 Table). The mRNA abundance of these transcripts at these three time-points after the $H_2O_2$ treatment followed a profile similar to that of the microarray dataset (S3 Table), thus validating the quality of our assay. The Pearson correlation test of the microarray against the qRT-PCR measurements yielded a correlation coefficient of $R^2 = 0.81$ (n = 4), suggesting that RNA-seq dataset correlated positively and tightly with the qRT-PCR quantification (Fig 2).

## A hypothetical gene plays some role in the oxidative stress response in *Xoc*

Sensing, detoxification, and adaptation to oxidative stress play a critical role during successful pathogen infection and pathogenesis by *Xanthomonas* [30]. The XOC_1643 (outer membrane channel protein), XOC_3249 (membrane protein YnfA) and XOC_2868 (hypothetical protein) and mutants clearly displayed a greater sensitivity to $H_2O_2$ than did the BLS256 wild-type and complemented strains (Fig 3). Not surprisingly, these three mutants showed decreased pathogenicity when compared with the wild-type (Fig 4). However, the XOC_1946 (TonB-dependent receptor) mutants showed a greater resistance to $H_2O_2$ but a decreased pathogenicity when compared with the wild-type (Fig 4). TonB-dependent receptor was proven to be involved in the transport of plant-derived molecules such as sucrose and maltodextrins in previous studies and yet delayed the disease symptom development in plants to some extent [31, 32]. Our studies show similar results with earlier researches, suggests that XOC_1946 is

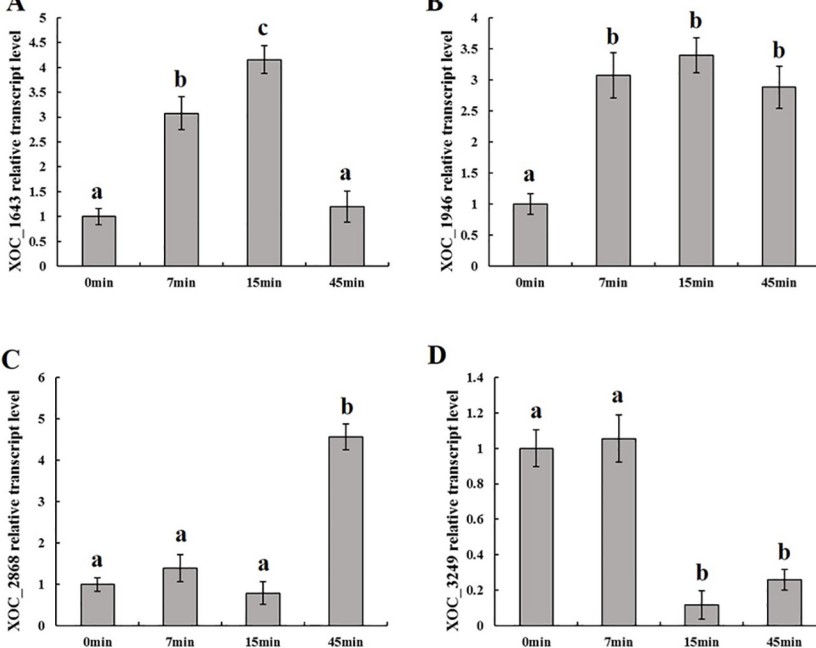

**Fig 2. Real-time quantitative PCR analysis.** Transcript levels of the four candidates under oxidative stress at 7, 15, and 45 min after $H_2O_2$ treatment. Values given are the means of five replicate measurements from a representative experiment. The experiment was repeated five times, and similar results were obtained. Columns with the same letters are not significantly different from each other by t-test (i.e., $P \geq 0.05$). Error bars represent the standard deviations.

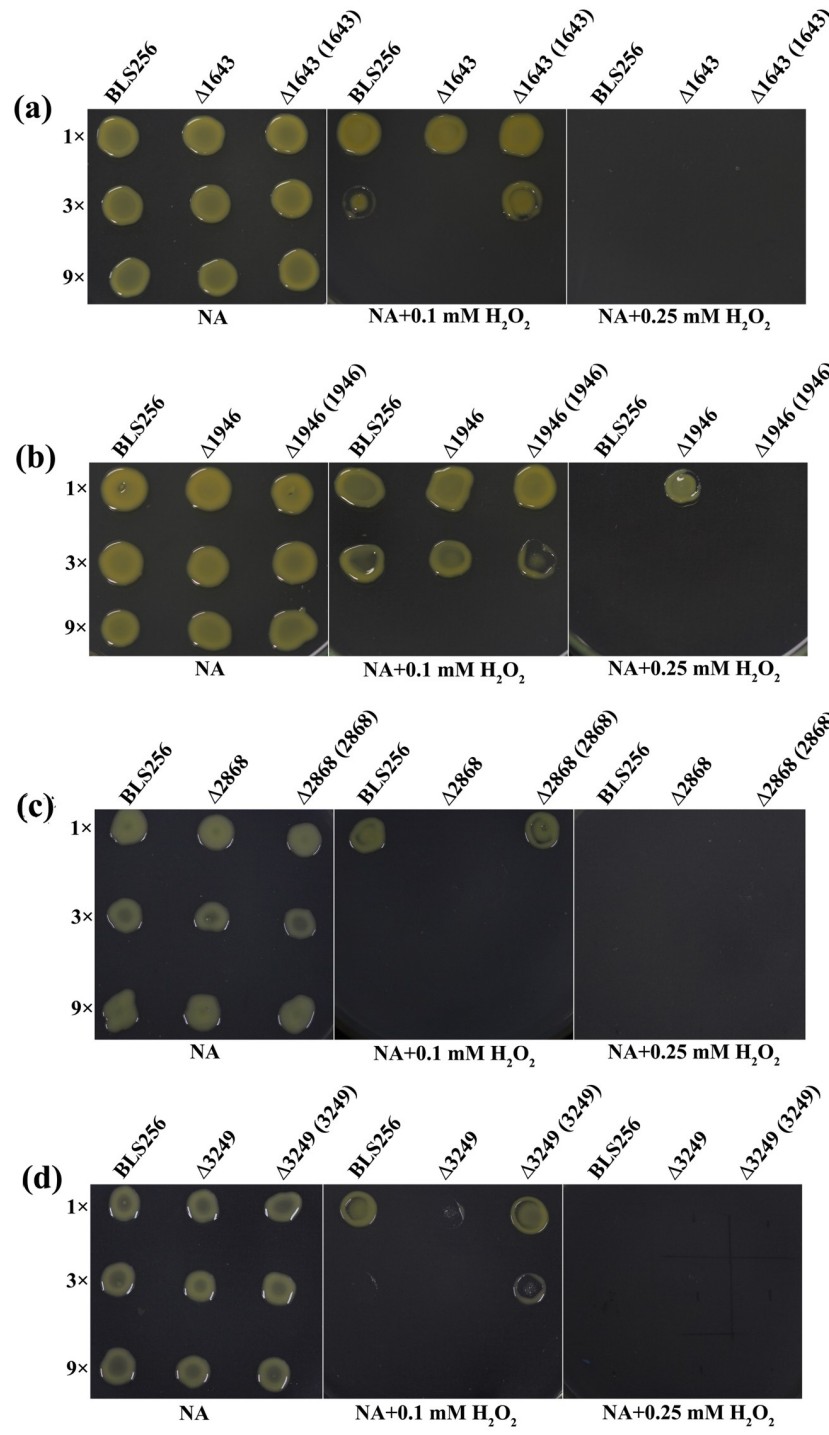

**Fig 3. Gene mutations changed the resistance to H$_2$O$_2$ in *Xanthomonas oryzae* pv. *oryzicola* (*Xoc*).** *Xoc* strains, including the wild-type strain BLS256, the gene deletion mutants Δ1643, Δ1946, Δ2868, and Δ3249, and their complemented strains Δ1643 (1643), Δ1946 (1946), Δ2868 (2868), and Δ3249 (3249), were grown on nutrient broth agar (NA) plates with 0 mM H$_2$O$_2$, 0.1 mM H$_2$O$_2$, or 0.25 mM H$_2$O$_2$. Three replicates for each treatment were used, and the experiment was repeated three times. "1x": original cultures; "3x": three-fold dilutions of cultures; "9x": nine-fold dilutions of cultures.

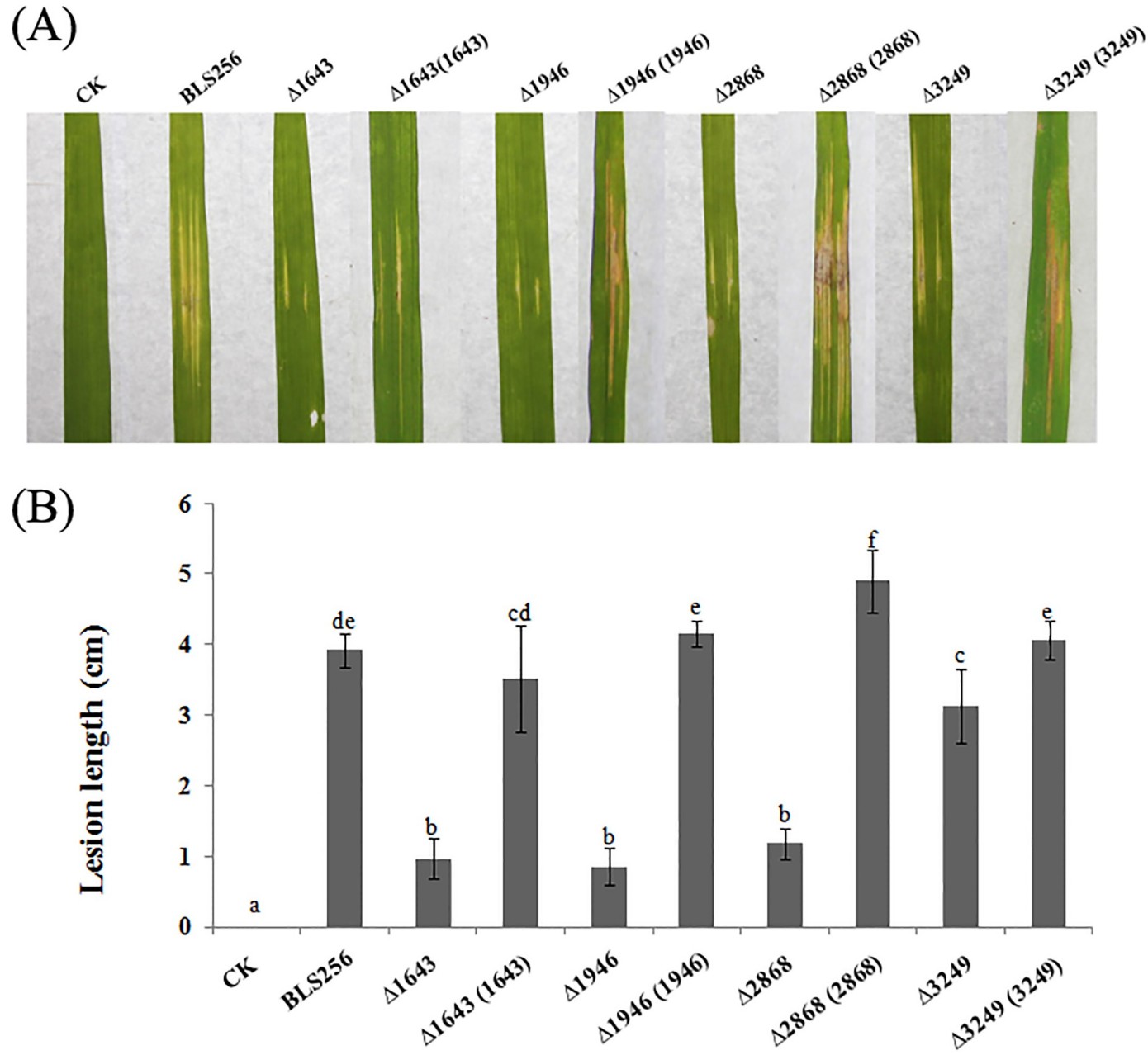

**Fig 4. Gene mutations changed the virulence and growth of *X. oryzae* pv. *oryzae in planta*.** (A) Symptoms induced by different *X. oryzae* pv. *oryzae* strains inoculated to leaves of 2-month-old rice plants (IR24, a susceptible cultivar). Photographs were taken at 14 dpi. Three replicates for each treatment were used, and the experiment was repeated three times. (B) Lesion lengths on the leaves of adult susceptible rice (cv. IR24, 2months old) inoculated with the *Xoc* strains. Different letters above the data bars indicate a significant difference between the wild-type strain and deletion mutants or complemented strains ($P < 0.05$; *t*-test).

involved in the transport of $H_2O_2$. As genes *xoc_1643*, *xoc_1946* and *xoc_3249* have proven relationships with an oxidative stress response, this result confirmed the accuracy of our RNA-seq result in this study. Interestingly, XOC_2868, a hypothetical protein, was also involved in the stress response. Sequence analysis revealed that this gene has a MntR domain, thus indicating its potential role in superoxide resistance regulation [33].

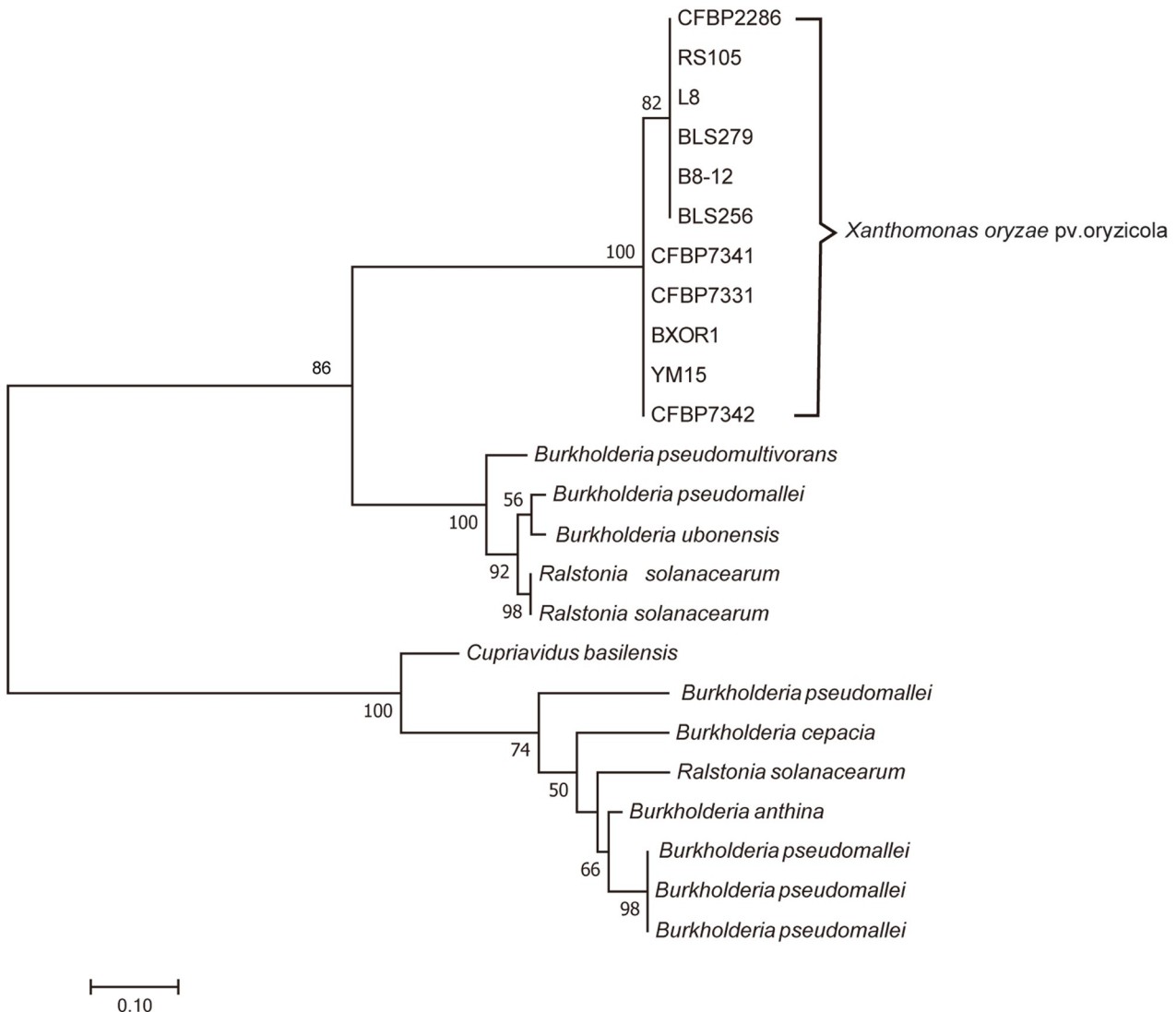

**Fig 5. A phylogeny of the *Xoc_2868*, a MntR-like gene.** The phylogenetic tree shown was calculated using the maximum likelihood (ML) program in PhyML [23]. Only the ML values ≥ 50% were shown. Bar = 0.1 substitution per site.

### *xoc_2868* is a horizontally transferred gene

A comprehensive BLAST sequence analysis revealed that this MntR-like gene exists widely in *Xoc* but not in other *Xanthomonas* species (Fig 5). Interestingly, homologs of this gene were found to present in many *Burkholderiaceae* family bacteria, such as *Burkholderia*, *Ralstonia*, and *Cupriavidus* (Fig 5). The phylogenetic analysis further suggested that this gene in *Xoc* might have originated from a transfer from a *Burkholderiaceae* ancestor over the course of evolutionary history, an inference with high bootstrapping support (Fig 5). Notably, many genes located adjacent to *Xoc_2868* are transposase (i.e., from *Xoc_2859* to *Xoc_2865*). Ochman et al. suggested that HGT is mediated by bacteriophage integrases or mobile element transposases, while Keeling et al. suggested that a phylogenetic tree is the gold standard by which to identify HGT [34, 35]. The results presented here strongly suggest that *xoc_2868* is a horizontally transferred gene.

## Discussion

In this study, we demonstrated, for the first time, that TonB-dependent receptors (TBDRs) is involved in resistance to $H_2O_2$ and virulence in *Xoc*. Indeed, the trigger of these TBDRs is reportedly involved in many different kinds of stress responses such as oxidative stress, iron stress and zinc stress [36]. Researchers also found that TBDRs are required for full virulence of *Xanthomonas campestris* pv. *campestris* to *Arabidopsis* [30]. We may infer that TBDRs may play some role in the oxidative response in *Xoc*. Hence, the function of one TBDRs gene *Xoc_1946* was confirmed by $H_2O_2$ resistance assays and pathogenicity assays. Our results show that *Xoc_1946* did play a role in oxidative response and pathogenicity. However, the role of the TBDRs in these bacterial species remains to be investigated.

There are many oxidative-associated gene clusters were found involved in DEGs on our study. Alkyl hydroperoxide reductase have been reported responsible for alky1 peroxide metabolism in *Xanthomonas*. It is the best characterized microbial enzyme involved in organis peroxide metabolism[30]. *suf* operon, which consists of a list of cysteine desulfurase encoding genes, is involved in the assembly of [Fe–S] clusters under oxidative stress, it is also known as being necessary for virulence of the plant pathogen *Erwinia chrysanthemi*[31]. Another operon *pst*, which has been found differentially expressed in this study, was also been considered to improve heat, oxygen and starvation stress resisitance in *Lactococcus lactis*[37]. Our results indicate that the role of these clusters in bacterial resistance to oxidative-stress might not be diverse in different bacterial species.

Prior research has demonstrated that $H_2O_2$ causes a slower rate of ribosomal run-off, while the expression of several ribosomal proteins associated with the translation of the stress response-associated genes was increased [38]. Our findings show that there is a gene cluster encoding ribosomal proteins were up-regulated, this result indicate that this gene cluster contributes to the translation of oxidative stress associated genes in our *Xoc* strain.

It is well-known that the oxidative stress generated in the plant response to pathogens will decrease the intracellular pH [38]. Although direct evidence is lacking for *Xanthomonas* species, the mutants of these genes from several other bacteria showed clear growth defects under low pH, thus indicating the F1F0 ATPase complex cluster which was found up-regulated in our study is important for maintain the ΔpH [39].

It is now widely appreciated that a time-course transcriptome analysis can help us to better understand how organisms react to stress conditions over time [40]. Here, we set three time points corresponding to the early, middle, and late response stages. Very few genes were significantly differentially expressed in the early stage (S3 Table). Interestingly, however, we did find that one gene encoding SoxR, a redox-sensitive transcriptional activator, was the highest up-regulated gene in this time stage. In *E. coli*, *soxR* and *soxS* were shown to control the superoxide response regulon of *E. coli* [41]. Since *Xoc* lacks the homolog of *soxS*, and *soxR* is the only transcriptional regulator, we may infer that this gene triggers the *Xoc* oxidative stress response.

## Conclusions

In this research, gene expressions of *Xoc* strain BLS256 in response to a time-series $H_2O_2$ treatment have been presented by RNA-Seq analysis. In general, 7, 177, and 246 genes were differentially regulated in the early, middle, and late stages, respectively. The *soxR* gene was highly up-regulated in the early stage, indicating that this gene triggers the *Xoc* oxidative stress response. In addition, the sensitivity to $H_2O_2$ and pathogenicity of four DEGs' mutants were investigated, and the results prove strong relationships between these DEGs and oxidative stress response. Interestingly, the results about a hypothetical protein XOC_2868 presented here strongly suggest that it is encoded by a horizontally transferred gene.

## Supporting information

**S1 Table. Strains and plasmids used in this study.**
(DOCX)

**S2 Table. Primers used in this study.**
(DOCX)

**S3 Table. Differential expressed genes in the early, middle and late stage.**
(XLSX)

**S4 Table. Gene clusters that were differential expressed in the early (red color), middle (yellow color) and late (blue color) stages.**
(XLSX)

## Author Contributions

**Data curation:** Yuan Fang, Haoye Wang, Bo Zhu.

**Formal analysis:** Haoye Wang, Xia Liu, Bo Zhu.

**Funding acquisition:** Yuan Fang, Bo Zhu.

**Investigation:** Haoye Wang.

**Methodology:** Yuan Fang, Haoye Wang, Xia Liu.

**Supervision:** Yuchun Rao.

**Writing – original draft:** Yuan Fang, Haoye Wang, Yuchun Rao.

**Writing – review & editing:** Yuan Fang, Dedong Xin, Yuchun Rao.

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
