## [Decision Letter · Decision Letter 0]

24 Jul 2019

PONE-D-19-16123

Transcriptome analysis of Xanthomonas oryzae pv. oryzicola exposed to H2O2 reveals horizontal gene transfer contributes to its oxidative stress response

PLOS ONE

Dear Mr. Rao,

Thank you for submitting your manuscript to PLOS ONE. After careful consideration, we feel that it has merit but does not fully meet PLOS ONE’s publication criteria as it currently stands. Therefore, we invite you to submit a revised version of the manuscript that addresses the points raised during the review process.

We would appreciate receiving your revised manuscript by Sep 07 2019 11:59PM. To enhance the reproducibility of your results, we recommend that if applicable you deposit your laboratory protocols in protocols.io, where a protocol can be assigned its own identifier (DOI) such that it can be cited independently in the future. For instructions see: http://journals.plos.org/plosone/s/submission-guidelines#loc-laboratory-protocols

We look forward to receiving your revised manuscript.

Kind regards,

Kandasamy Ulaganathan

Academic Editor

PLOS ONE

Journal Requirements:

2) We note that you have included the phrase “data not shown” in your manuscript. Unfortunately, this does not meet our data sharing requirements. PLOS does not permit references to inaccessible data. We require that authors provide all relevant data within the paper, Supporting Information files, or in an acceptable, public repository. Please add a citation to support this phrase or upload the data that corresponds with these findings to a stable repository (such as Figshare or Dryad) and provide and URLs, DOIs, or accession numbers that may be used to access these data. Or, if the data are not a core part of the research being presented in your study, we ask that you remove the phrase that refers to these data.

3) Please amend the manuscript submission data (via Edit Submission) to include author Xin Dedong.

Reviewers' comments:

Reviewer's Responses to Questions

**Comments to the Author**

1. Is the manuscript technically sound, and do the data support the conclusions?

Reviewer #1: Partly

Reviewer #2: Partly

2. Has the statistical analysis been performed appropriately and rigorously? 

Reviewer #1: No

Reviewer #2: Yes

3. Have the authors made all data underlying the findings in their manuscript fully available?

Reviewer #1: No

Reviewer #2: Yes

4. Is the manuscript presented in an intelligible fashion and written in standard English?

Reviewer #1: No

Reviewer #2: No

5. Review Comments to the Author

Reviewer #1: This manuscript described a time-resolved RNA-seq result that reveals Xoc transcriptome change related to oxidative stress response during exposed to hydrogen peroxide. Following the differentially expressed gene analysis, further investigations were carried out to determine 4 DEGs’ role in terms of antioxidation and virulence. This study provided useful information to related professional researcher. However, according to this manuscript, it is regretful to learn that part of result could not sufficient support the conclusion made by the authors.

Major issues

1. The virulence assay result could not support the conclusion made by the authors.

According to Fig 4, I assumed the authors using leaf piercing method to inoculate Xoc on rice leaf. But the authors did not clearly describe this in manuscript, the inoculate process was missing. By using leaf piercing method, the virulence should be evaluated by either calculating the means of lesion length (n >= 10), or establishing bacterial growth curves, or both. But in Fig 4, authors only showed a typical lesion caused by leaf piercing inoculation for each Xoc strain, which is not enough to represent their virulence.

2. To demonstrate whether a gene is required for full virulence of Xoc to rice, the virulence mutant should be assayed along with its complementation strain, which have not been showed in this manuscript.

3. The GEO data is not accessible. The authors declared all raw RNA-seq data is available on GEO at line 315-318. But the given accession numbers are invalid.

Minor issues

1. The organization of Introduction part is poor. And I suggest authors describe some known oxidative stress response process in microbe as part of background.

2. Some bioinformatic methods mentioned in Materials and Methods need proper citation. e.g. in line 135, the TMM method, and in line 155, the 2(-ΔΔC(T)) method, both need a proper citation.

3. At line 101, author mentioned “Cultures were treated with 0.1mM H2O2 in a 28 ℃ shaking incubator.” without describing how to conduct H2O2 treatment. I assumed the authors were closely following the method in Upadhya at.el work. I recommended the authors to provide more details in this part to make it more readable.

4. At line 146, the total RNA extraction process is missing.

5. At line 182, more details for how to interduce H2O2 into NB agar plates is required.

6. At line 190 to 193. author mentioned “Plant phenotypes were scored 24 h post-inoculation for the HR in tobacco, 3 days post-inoculation (dpi) for water-soaked symptoms, and 14 dpi for lesion length.” But there is not any hypersensitive reaction or water-soaked symptom result in this manuscript. Please make sure only the involved methods were included in this manuscript.

7. At line 203 to 206, the result indicated there were about 75% clear reads mapped on ribosomal RNA. Is this a typical rRNA/mRNA ratio for RiboMinus treated sample?

8. At line 235, authors should provide the full form of GO and KEGG.

9. At line 266, what’s the mean of “encompassed a range of expression levels at 7, 15, and 45 minutes”?

10. At line 267-272, author mentioned a microarray dataset but never described it. I assume author intend to refer the RNA-seq data in this study.

11. In Fig 3, the authors did not describe the meaning of captions “1x”, “3x”, and “9x”.

12. At line 315-318, the authors should not put the accession numbers in the main part of manuscript. The authors should list all accession numbers instead of an accession number range.

13. At line 347-358, the phylogenetic analysis result should be moved to the Result section.

14. The order of supporting information label incorrect.

15. The S1 Table, titled as “Differential expressed genes in the early, middle and late stage.”, missing heading at column one.

16. The S4 Table missing its heading.

Overall, the writing of this manuscript is very poor. The authors should proof the writing of the manuscript seriously, or consider hiring a professional copyeditor to handle it.

Reviewer #2: In the manuscript ‘Transcriptome analysis of Xanthomonas oryzae pv. oryzicola exposed to H2O2 reveals horizontal gene transfer contributes to its oxidative stress response’ the authors study the gene expressions of XoC strain BLS256 in response to time series H2O2 treatment, following RNA-Seq analysis. They claim in the Abstract, that ‘soxR triggers and regulates XoC oxidative stress response in early infection, gene expression kinetics among the time-series samples namely for TonB-dependent receptors and suf and pst operons. However, there is no clear data in the Results to corroborate these claims.

The finding that a MntR-like hypothetical protein is highly upregulated at a later time-point and that it is likely to be a horizontally transferred gene is interesting.

However, the English and the overall presentation of the manuscript needs to be improved considerably.

The descriptions of the Supporting data do not match the actual Supporting Tables. Hence,

In lines 249, 261 and 340: S3 Table needs to be replaced by S1 Table.

The authors have validated the role of 4 selected genes for their role in H2O2 sensitivity and pathogenicity by mutant generation and their complementation. It might be better to perform the experiments with at least two-independently generated mutants for each gene.

6. PLOS authors have the option to publish the peer review history of their article (what does this mean?). If published, this will include your full peer review and any attached files.

Reviewer #1: No

Reviewer #2: No

---

## [Author Response · Author response to Decision Letter 0]

6 Sep 2019

Response: we have checked our manuscript carefully, it meets PLOS ONE's style requirements.

2) We note that you have included the phrase “data not shown” in your manuscript. Unfortunately, this does not meet our data sharing requirements. PLOS does not permit references to inaccessible data. We require that authors provide all relevant data within the paper, Supporting Information files, or in an acceptable, public repository. Please add a citation to support this phrase or upload the data that corresponds with these findings to a stable repository (such as Figshare or Dryad) and provide and URLs, DOIs, or accession numbers that may be used to access these data. Or, if the data are not a core part of the research being presented in your study, we ask that you remove the phrase that refers to these data.

Response: Thank you for your suggestion, the phrase “data not shown” has been removed in revised manusscript. 

3) Please amend the manuscript submission data (via Edit Submission) to include author Xin Dedong.

Response: Sorry for the mistake, the information of author Xin Dedong will be added when resubmitting.

Reviewer #1:

Major issues

1. The virulence assay result could not support the conclusion made by the authors.

According to Fig 4, I assumed the authors using leaf piercing method to inoculate Xoc on rice leaf. But the authors did not clearly describe this in manuscript, the inoculate process was missing. By using leaf piercing method, the virulence should be evaluated by either calculating the means of lesion length (n >= 10), or establishing bacterial growth curves, or both. But in Fig 4, authors only showed a typical lesion caused by leaf piercing inoculation for each Xoc strain, which is not enough to represent their virulence.

Response: We thank the reviewer for the suggestions. We did inoculate plants by leaf piercing. Sorry for the negligence. In the revised manuscript, The inoculation process was presented. Which is “Briefly, Xanthomonas cells were grown in NB broth at 28°C and 200 rpm for 16 h, when cells approached the exponential phase of growth. Bacterial cells were then harvested by centrifugation, washed twice, and resuspended in sterile water to an optical density at 600 nm (OD600) of 0.3 (approximately 1�108 CFU/mL). Bacteria were inoculated into leaves of adult rice plants (Oryza sativa cv. IR24, susceptible to Xoc, 2 months old) using leaf piercing for lesion length measurement for evaluating water-soaked symptoms.”

We realized there was not enough to represent the virulence only by a typical lesion in Fig 4, so we add the results of the lesion length to evaluate the virulence together with the symptoms of leaf lesion (which is shown in Fig 4).

2. To demonstrate whether a gene is required for full virulence of Xoc to rice, the virulence mutant should be assayed along with its complementation strain, which have not been showed in this manuscript.

Response: Thank you for the excellent question. Actually we have tested the virulence of these complementation strains. We have put the results in our revised manuscript and Fig 4.

3. The GEO data is not accessible. The authors declared all raw RNA-seq data is available on GEO at line 315-318. But the given accession numbers are invalid.

Response: Sorry for the mistake. We have corrected this in the new version. “All the RNA-seq data has been deposited in BioProject database (https://www.ncbi.nlm.nih.gov/bioproject) and the accession number is PRJNA350867.”

Minor issues

1. The organization of Introduction part is poor. And I suggest authors describe some known oxidative stress response process in microbe as part of background.

Response: Great suggestion. As suggested, we reorganized the order of Introduction part. Xanthomonas was introduced at first, then we descibe the oxidative stress response of microorganisms. Subsequently, RNA-seq and BLS256 were briefly introduced. At last, the purpose of our study was clearly presented.

Also, we add a part of description on oxidative stress response of microorganisms. “Oxidative burst is a process in which high concentrations of reactive oxygen species (ROS) are produced at the plasma membrane in the vicinity of a pathogen. This activity in plants may directly kill the pathogen, slows its growth by producing toxins, or even act as a signaling cascade leading to various defenses, including the hypersensitive response (HR) and cell wall modifications. Since ROS activity is a common feature of plant defense systems and the mechanism involved in pathogen cell death, any pathogen that is able to resist this effect is likely privileged. These mechanisms require complex and well-orchestrated reactions involving both radical scavenging and enzymatic repair activities. The cellular response to oxidative stress in the model organism, Escherichia coli, has been largely elucidated. When E. coli cells are treated with a low dose of H2O2, growth arrest occurs immediately and the expression of approximately 30 genes is maximally induced within 10–30 minutes of treatment. Among them, the oxidized form of the transcriptional regulator, OxyR, induces many genes such as katG (encoding catalase G), ahpCF (encoding alkyl hydroperoxide reductase), and trxC (encoding reduced thioredoxin) to remove intracellular H2O2, maintain redox homeostasis and ultimately enable cells to resume growth. These studies demonstate that the genes involved in oxidative-stress response are vital for bacteria survival.”

2. Some bioinformatic methods mentioned in Materials and Methods need proper citation. e.g. in line 135, the TMM method, and in line 155, the 2(-ΔΔC(T)) method, both need a proper citation.

Response: Thank you for your suggestion. The proper citation have been added in revised manuscript. 

3. At line 101, author mentioned “Cultures were treated with 0.1mM H2O2 in a 28 ℃ shaking incubator.” without describing how to conduct H2O2 treatment. I assumed the authors were closely following the method in Upadhya at.el work. I recommended the authors to provide more details in this part to make it more readable.

Response: Thank you for your suggestion. I have made some modifications in revised manuscript. “Cultures were treated with 0.1mM H2O2 (Fluka) and constantly stirred at 330 rpm in a 28 ℃ shaking incubator.”

4. At line 146, the total RNA extraction process is missing.

Response: Sorry for the mistake. “Total RNA immediately extracted using fthe RNeasy Protect Bacteria Mini Kit (QIAGEN) protocal.”

1. At line 182, more details for how to interduce H2O2 into NB agar plates is equired.

Response: Thank you for your suggestion. The assay plates were prepared by adding H2O2 to the sterilized NB medium in concentrations of 0, 0.1 and 0.25 mM, respectively.

6. At line 190 to 193. author mentioned “Plant phenotypes were scored 24 h post-inoculation for the HR in tobacco, 3 days post-inoculation (dpi) for water-soaked symptoms, and 14 dpi for lesion length.” But there is not any hypersensitive reaction or water-soaked symptom result in this manuscript. Please make sure only the involved methods were included in this manuscript.

Response: Sorry for the mistake. There is no hypersensitive reaction in our study, we have correted this sentences.

7. At line 203 to 206, the result indicated there were about 75% clear reads mapped on ribosomal RNA. Is this a typical rRNA/mRNA ratio for RiboMinus treated sample?

Response: Great Question. It’s not a typical rRNA/mRNA ratio for RiboMinus treated sample. Xanthomonas oryzae pv. oryzicola is not a regular bacteria that in the recommended list of the RiboMinus kit. So the effect of removing is not so good. 

8. At line 235, authors should provide the full form of GO and KEGG.

Response: Sorry for the typo. The full form of GO and KEGG have been added in revised manuscript. There are “Gene Ontolog (GO) and Kyoto Encyclopedia of Genes and Genomes (KEGG)”.

9. At line 266, what’s the mean of “encompassed a range of expression levels at 7, 15, and 45 minutes”?

Response: Thank you for your question. It is really an ambiguous writing. We have correct this sentence by “showed different levels of expression at 7, 15, and 45 minutes”.

10. At line 267-272, author mentioned a microarray dataset but never described it. I assume author intend to refer the RNA-seq data in this study.

Response: That’s right. It refered to the S3 Table here.

11. In Fig 3, the authors did not describe the meaning of captions “1x”, “3x”, and “9x”.

Response: Sorry for the negligence. I have added the explanations at the end of the Fig 3 legends. “1x”: original cultures; “3x”: three-fold dilutions of cultures; “9x”: nine-fold dilutions of cultures.

12. At line 315-318, the authors should not put the accession numbers in the main part of manuscript. The authors should list all accession numbers instead of an accession number range.

Response: Thank you for your suggestion. I have correct the accession number and put this sentences in the chapter of Resluts.

13. At line 347-358, the phylogenetic analysis result should be moved to the Result section.

Response: Thank you for your suggestion. The phylogenetic analysis has been moved to the Results section. 

14. The order of supporting information label incorrect.

Response: Sorry for the mistake. We have placed them in correct order in the revised manuscript.

15. The S1 Table, titled as “Differential expressed genes in the early, middle and late stage.” missing heading at column one.

Response: Sorry for the mistake. I have added the heading in column one.

16. The S4 Table missing its heading.

Response: Sorry for the mistake. I have added the heading in column one.

Overall, the writing of this manuscript is very poor. The authors should proof the writing of the manuscript seriously, or consider hiring a professional copyeditor to handle it.

Response: We have checked the grammar of the manuscript carefully and the English in the revised manuscript has been edited and improved by a native English speaker. We hope it will reach the standards of the journal for publication.

Reviewer #2: 

In the manuscript ‘Transcriptome analysis of Xanthomonas oryzae pv. oryzicola exposed to H2O2 reveals horizontal gene transfer contributes to its oxidative stress response’, the authors study the gene expressions of Xoc strain BLS256 in response to time series H2O2 treatment, following RNA-Seq analysis. They claim in the Abstract, that ‘soxR triggers and regulates Xoc oxidative stress response in early infection, gene expression kinetics among the time-series samples namely for TonB-dependent receptors and suf and pst operons’. However, there is no clear data in the Results to corroborate these claims.

Response: Thank you for your good suggestion. It is a hypothesis that ‘soxR triggers and regulates Xoc oxidative stress response in early infection, gene expression kinetics among the time-series samples namely for TonB-dependent receptors and suf and pst operons’. It will be confirmed in subsequent research. So I have rewrited the Abstract in revised manuscript, this sentence has been deleted.

The finding that a MntR-like hypothetical protein is highly upregulated at a later time-point and that it is likely to be a horizontally transferred gene is interesting.

However, the English and the overall presentation of the manuscript needs to be improved considerably.

Response: We have checked the grammar of the manuscript carefully and the English in the revised manuscript has been edited and improved by a native English speaker. We hope it will reach the standards of the journal for publication.

The descriptions of the Supporting data do not match the actual Supporting Tables. Hence, In lines 249, 261 and 340: S3 Table needs to be replaced by S1 Table.

Response: Sorry for the mistake. I have corrected it in revised manuscript.

The authors have validated the role of 4 selected genes for their role in H2O2 sensitivity and pathogenicity by mutant generation and their complementation. It might be better to perform the experiments with at least two-independently generated mutants for each gene.

Response: Thank you for your good suggestion. Actually, we got at least five mutants for each gene, the results they presented were consistent.

---

## [Decision Letter · Decision Letter 1]

23 Sep 2019

Transcriptome analysis of Xanthomonas oryzae pv. oryzicola exposed to H2O2 reveals horizontal gene transfer contributes to its oxidative stress response

PONE-D-19-16123R1

Dear Dr. Rao,

We are pleased to inform you that your manuscript has been judged scientifically suitable for publication and will be formally accepted for publication once it complies with all outstanding technical requirements.

With kind regards,

Kandasamy Ulaganathan

Academic Editor

PLOS ONE

Additional Editor Comments (optional):

Please carry out the minor corrections pointed by the Reviewer

Reviewers' comments:

Reviewer's Responses to Questions

**Comments to the Author**

1. If the authors have adequately addressed your comments raised in a previous round of review and you feel that this manuscript is now acceptable for publication, you may indicate that here to bypass the “Comments to the Author” section, enter your conflict of interest statement in the “Confidential to Editor” section, and submit your "Accept" recommendation.

Reviewer #2: All comments have been addressed

2. Is the manuscript technically sound, and do the data support the conclusions?

Reviewer #2: Yes

3. Has the statistical analysis been performed appropriately and rigorously? 

Reviewer #2: Yes

4. Have the authors made all data underlying the findings in their manuscript fully available?

Reviewer #2: Yes

5. Is the manuscript presented in an intelligible fashion and written in standard English?

Reviewer #2: Yes

6. Review Comments to the Author

Reviewer #2: The manuscript looks considerably better and improved.

Following are some minor corrections:

Line 70: limited instead of limite

line 71: space between elucidate and the

line 105: delete Cell

line 106: delete was

line 133: space between having and the

line 276: remove additional space between different and levels

line 362: organic instead of organis

7. PLOS authors have the option to publish the peer review history of their article (what does this mean?). If published, this will include your full peer review and any attached files.

Reviewer #2: No

---

## [Editor Report · Acceptance letter]

24 Sep 2019

PONE-D-19-16123R1 

Transcriptome analysis of *Xanthomonas oryzae* pv.  *oryzicola*  exposed to H2O2 reveals horizontal gene transfer contributes to its oxidative stress response 

Dear Dr. Rao:

I am pleased to inform you that your manuscript has been deemed suitable for publication in PLOS ONE. Congratulations! Your manuscript is now with our production department. 

With kind regards,

on behalf of

Dr. Kandasamy Ulaganathan 

Academic Editor

PLOS ONE